# Traces of Canine Inflammatory Bowel Disease Reflected by Intestinal Organoids

**DOI:** 10.3390/ijms25010576

**Published:** 2024-01-01

**Authors:** Barbara Pratscher, Benno Kuropka, Georg Csukovich, Pavlos G. Doulidis, Katrin Spirk, Nina Kramer, Patricia Freund, Alexandro Rodríguez-Rojas, Iwan A. Burgener

**Affiliations:** 1Clinic for Small Animals, Division for Small Animal Internal Medicine, Department for Small Animal and Horses, University of Veterinary Medicine, 1210 Vienna, Austria; barbara.pratscher@vetmeduni.ac.at (B.P.); georg.csukovich@vetmeduni.ac.at (G.C.); pavlos.doulidis@vetmeduni.ac.at (P.G.D.); katrin.spirk@vetmeduni.ac.at (K.S.); patricia.freund@vetmeduni.ac.at (P.F.); 2Institute of Chemistry and Biochemistry, Freie Universität Berlin, 14195 Berlin, Germany; kuropka@zedat.fu-berlin.de

**Keywords:** canine IBD, IBD, inflammatory bowel diseases, organoid, chronic enteropathy, disease modeling

## Abstract

Inflammatory bowel disease (IBD) is a chronic inflammatory condition that affects humans and several domestic animal species, including cats and dogs. In this study, we have analyzed duodenal organoids derived from canine IBD patients using quantitative proteomics. Our objective was to investigate whether these organoids show phenotypic traits of the disease compared with control organoids obtained from healthy donors. To this aim, IBD and control organoids were subjected to quantitative proteomics analysis via liquid chromatography–mass spectrometry. The obtained data revealed notable differences between the two groups. The IBD organoids exhibited several alterations at the levels of multiple proteins that are consistent with some known IBD alterations. The observed phenotype in the IBD organoids to some degree mirrors the corresponding intestinal condition, rendering them a compelling approach for investigating the disease and advancing drug exploration. Additionally, our study revealed similarities to some human IBD biomarkers, further emphasizing the translational and comparative value of dogs for future investigations related to the causes and treatment of IBD. Relevant proteins such as CALU, FLNA, MSN and HMGA2, which are related to intestinal diseases, were all upregulated in the IBD duodenal organoids. At the same time, other proteins such as intestinal keratins and the mucosal immunity PIGR were depleted in these IBD organoids. Based on these findings, we propose that these organoids could serve as a valuable tool for evaluating the efficacy of therapeutic interventions against canine IBD.

## 1. Introduction

Inflammatory bowel disease (IBD) is a chronic condition that affects the gastrointestinal tract and is a serious threat to global health. IBD already affects nearly seven million people worldwide and its prevalence is constantly increasing. The wide-ranging effects of this expanding load include enormous social and economic demands on governments and healthcare systems. In order to address the effects of this complicated disease on individuals and society as a whole, there is an urgent need for better understanding, efficient management options and increased support [1].

IBD extends beyond humans; it can also impact dogs and cats, exhibiting both distinctive traits and resemblances to human conditions such as Crohn’s disease and ulcerative colitis [2]. This multifaceted disorder involves intricate pathogenic processes and is marked by chronic inflammatory responses within the gastrointestinal tract. The immune system becomes dysregulated, leading to recurring cycles of inflammation. Additionally, IBD profoundly disrupts the microbiome and general metabolism, impairing the absorption of nutrients and water. This, in turn, manifests as symptoms such as diarrhea, weight loss and abdominal pain. The intricate interplay of factors, which may encompass genetic susceptibility, pathogens, the microbiome, environmental toxins and allergies, highlights the complexity of IBD. This underscores the significance of implementing comprehensive management strategies for affected animals [3].

The exact cause of IBD is still unknown. However, given that some dog breeds are more commonly affected than others, a genetic predisposition may have a role in the development of the illness [4]. Genetic factors have also been identified as a significant parameter in humans [5]. Furthermore, additional triggers such as microbial infections and allergens may contribute to the disease. These numerous factors work together, explaining the high complexity of the inflammatory response in IBD. More research is necessary to completely understand the mechanisms underlying IBD onset and progression [6].

To gain a better understanding of the etiology of IBD, innovative techniques such as transcriptome sequencing and proteomics have emerged as valuable tools. Transcriptome sequencing allows for a comprehensive analysis of the entire set of RNA molecules present in a cell or tissue, providing insights into the gene expression patterns and regulatory mechanisms involved in IBD. By comparing the transcriptomes of healthy individuals with those from IBD patients, we can identify differentially expressed genes and unravel the key molecular pathways implicated in the disease [7].

In addition to transcriptome sequencing, proteomics has emerged as an important tool in IBD research. Proteomic approaches enable the large-scale identification and quantification of proteins within a given biological sample. By employing mass spectrometry, researchers can dissect the proteome of IBD patients and uncover disease-specific alterations in protein expression, post-translational modifications and protein–protein interactions. This deeper understanding of the proteomic landscape of IBD offers valuable insights into pathobiological mechanisms and can help identify potential therapeutic targets [8]. The implementation of proteomic techniques has initiated a revolution in biomarker discovery. In the field of clinical proteomics, several quantitative proteomic methodologies have emerged as efficient approaches.

Significant advancements in pluripotent stem cell technology and primary tissue culture methods have paved the way towards the three-dimensional culture of intestinal epithelial cells that self-assemble into “intestinal organoids”. These organoids represent a breakthrough in creating a novel, specific tool for studying gastrointestinal disorders. By faithfully recapitulating the complex architecture and cellular interactions of the intestinal tissue, these organoids provide a valuable platform to investigate the mechanisms underlying various disorders [9,10]. Therefore, intestinal organoids have been increasingly used over the past few years in gastrointestinal disease modelling. As a more sophisticated system than traditional two-dimensional cell culture, organoids also allow for the long-term maintenance and differentiation of a wide variety of cell types in a single dish due to their three-dimensional structure. Despite their complexity, intestinal organoids have the advantage that they consist of only one layer of epithelial cells, thus placing the intestinal epithelial lining at the center of the investigation. Therefore, intestinal organoids are useful tools for the study of a variety of complex disorders, including IBD [11].

In previous work, we established canine intestinal organoids from the small and large intestines [12]. Since the duodenum is one of the most affected sections of the intestine in canine IBD [13,14], we hypothesize that organoids originating from IBD patients retain disease traits that could lead to the identification of novel canine IBD biomarkers and therapeutic targets, thus constituting a valuable disease model. To test this hypothesis, our primary objective was to characterize duodenal canine organoids derived from individuals with IBD versus healthy control animals. We conducted a comprehensive molecular biological analysis to compare the physiological characteristics of IBD and control organoids. By examining the extent to which the organoids from IBD donors retained and replicated disease-associated traits by proteome analysis, we aimed at gaining new insights into the pathogenesis of IBD. Furthermore, we addressed the potential of these organoids as a valuable proxy for exploring therapeutic interventions. This research holds promise for advancing our understanding of IBD and may contribute to the discovery of new biomarkers and the development of targeted therapeutics.

## 2. Results and Discussion

Previous studies, including our own, have reported the successful generation of canine intestinal organoids, demonstrating their remarkable ability to mimic and recapitulate the key physiological characteristics the intestine [12,15,16]. These organoids faithfully represent the diverse cell types that make up the intestinal epithelium. By culturing these organoids in vitro, we observed the formation of three-dimensional structures that resemble the intricate architecture of the intestinal tissue [12,17].

We have successfully developed 3D duodenal organoids from the intestinal tissue of three dogs affected by IBD. The crypts were carefully collected from the duodenal segment, cultured and differentiated following a previously established protocol [12,18]. A preliminary study was carried out to characterize the redox biology of IBD-related organoids compared with non-IBD equivalents [18].

We next performed in-depth proteomics analysis using LC-MS combined with label-free quantification (Figure 1A). Initially, we identified 3214 proteins to be differentially expressed. Subsequently, proteins that were considered as contaminants (according to common contaminant database for MS) or were not identified in at least two out of the three replicates for at least one condition were excluded from the quantification. This filtering process resulted in the identification and quantification of 2735 proteins across all samples (see raw data in Appendix A).

A general overview of our proteomic dataset employing a principal component analysis (PCA) shows that the three IBD organoids clearly segregate from the controls in terms of global protein abundance and their pattern of expression (Figure 1B). Principal component analysis stands as a widely accepted mathematical technique that is employed across various disciplines for data analysis and modeling [19,20,21,22,23,24]. In the context of our study, PCA utilizes an orthogonal transformation to convert observations of potentially correlated variables—in this study, protein identification and relative protein intensities—into a series of values associated with linearly uncorrelated variables known as principal components (PC). These principal components serve as insightful representations, unveiling the internal structure of the data by capturing and elucidating its variance in an optimal manner [24]. The Perseus software incorporates principal component analysis (PCA) based on singular value decomposition (SVD) [25], a computational approach well suited for high-dimensional data. PCA within Perseus identifies the primary effects in the data and reveals the proteins responsible for the separation of different proteomic states [25]. In line with our hypothesis, the PCA analysis indicates that IBD organoids display an aberrant protein expression profile compared with control organoids originating from healthy individuals.

Using only the subset of proteins that showed a specific change in their relative abundance between IBD-organoid samples and control samples, we performed gene ontology enrichment analysis to investigate the functional significance of differentially expressed genes (DEGs) in the IBD organoids compared with the control organoids. By comparing DEGs against a background set of genes, we could identify gene ontology terms that provided valuable information about the protein classes enriched in IBD organoids (Figure 2). For this analysis, we mainly focused on DEGs that were consistently deregulated in the IBD organoids from the three independent donors.

Overall, various deregulation events were noted in IBD organoids, involving, e.g., proteins of the DNA metabolism, cytoskeleton proteins, metabolic enzymes, structural proteins and extracellular matrix proteins. Upregulated proteins were related to DNA metabolism and cell junctions, whereas cell adhesion molecules and defense/immunity protein classes were only present in the downregulated fraction.

Volcano plots depicting the relative protein intensities between the three IBD-organoid samples against the control sample are presented in Figure 3. Proteins showing significant changes in their relative abundance are highlighted if they show at least a 2-fold change in their relative intensity and their FDR-adjusted *p*-values is below 0.05. We detected a series of proteins in the upregulated fraction that were confirmed as being involved in intestinal function and several diseases. Among these proteins, CALU, FLNA, MSN and HMGA2 can be mentioned.

On the other hand, IBD-derived organoids exhibited a significant downregulation of keratins, particularly KRT18 and KRT86. Similarly, downregulation of PIGR was noted. Other proteins, including LAMA1, LAMC1 and DSP, were likewise expressed at lower levels in IBD organoids compared with controls.

This study aimed to advance our understanding of canine IBD by successfully generating canine duodenal organoids derived from canine donors diagnosed with this condition. Our primary aim was to scrutinize whether these organoids could preserve the phenotypic traits of IBD when compared with control organoids originating from healthy canine donors. We found that the difference between these organoids can be satisfactorily unraveled via state-of-the-art LC-MS analysis combined with label-free quantification. The results uncovered striking disparities between the IBD and control organoids, revealing pronounced physiological dysfunctions underpinned by altered levels of multiple proteins that could be closely associated with IBD pathogenesis. Considering these compelling findings, these canine IBD organoids hold great potential as a pivotal modeling system to study canine IBD.

In our dataset, we found a differentially decreased expression of PIGR (polymeric immunoglobulin receptor) independently in the three IBD organoids. This issue could potentially contribute to the development of intestinal immunodepression. PIGR has a crucial role in the mucosal immune system. It is required for the transport of polymeric IgA and IgM from the basolateral to the luminal side of the mucosal epithelium and the formation and release of secretory Ig, notably SIgA. This protease-resistant IgA complex is essential for maintaining the integrity of the mucosal immune barrier and defending against pathogenic microbes [27].

A decrease in PIGR expression compromises this surveillance function, potentially allowing the unchecked proliferation of pathogenic bacteria and a heightened inflammatory response in IBD. Another risk is the alteration of microbiota composition, since the reduction in PIGR expression may disrupt the balance of the gut microbiota, as polymeric SIgA antibodies have a major role in shaping the microbiome’s composition [28]. Dysbiosis of the gut microbiota has been implicated in the pathogenesis of IBD [29]. If the downregulation of PIGR in IBD can be confirmed for a larger number of canine IBD patients in vivo or ex vivo, future therapeutic interventions should focus on the reestablishment of PIGR expression and SIgA release to restore mucosal immunity and treat IBD.

A second important finding of this work is the decrease in the expression of keratins in the IBD organoids. A consequence of decreased intestinal keratins in the context of IBD is the compromised protection of the intestinal lining. Intestinal keratins are structural proteins that have a crucial role in maintaining the integrity and barrier function of the intestinal epithelium [30,31]. When their expression is reduced or altered in IBD, several significant implications could arise, including the loss of epithelial barrier integrity, since intestinal keratins, particularly KRT-8 and KRT-18, contribute to the structural stability of intestinal epithelial cells. A decrease in the levels of these keratins can weaken the tight junctions between epithelial cells, leading to increased intestinal barrier permeability [32]. This lack of keratins would, in turn, increase vulnerability to inflammation and impaired protection against luminal factors such as digestive enzymes and toxins. There is also a possibility of tissue damage and ulcerations and an alteration in cell signaling due to the role of intestinal keratins in some pathways that regulate inflammation and cell survival [30].

The CALU protein (calumenin) is upregulated in the IBD organoids; this protein has emerged as a critical biomolecule with dual roles in cancer and IBD, where it serves as an upregulated proinflammatory marker [33,34]. This multifunctional protein was originally identified due to its role in calcium binding and regulation, particularly in the endoplasmic reticulum. However, recent research has shed light on CALU’s involvement in inflammatory processes and its potential significance in disease pathogenesis. In the context of cancer, CALU has attracted attention as an upregulated proinflammatory marker in various malignancies. Elevated CALU expression has been associated with tumor progression and metastasis in several cancer types, including breast, lung and colorectal cancers. CALU’s proinflammatory role is linked to its involvement in modulating calcium-dependent signaling pathways, such as the activation of NF-κB, which promotes the transcription of proinflammatory genes. This upregulation of CALU in cancer contributes to a proinflammatory microenvironment within the tumor, fostering immune cell infiltration, angiogenesis and tumor growth [35].

Filamin A (FLNA) was also one of the most upregulated proteins in all IBD organoids. FLNA has a crucial role in the development and maintenance of the intestine. It is a large cytoplasmic protein that functions as an actin-binding scaffold, and its presence is essential for various cellular processes in the context of the intestine. FLNA acts as a co-regulator of cellular structure and motility. It cross-links actin filaments, the primary components of the cell’s cytoskeleton, providing strength and stability to the cells. This is crucial for the intestinal epithelium’s integrity, preventing damage and enhancing motility during processes such as peristalsis, the coordinated muscle contractions that move food through the digestive tract. FLNA promotes the formation of adherent junctions. These connections prevent the cells from becoming detached during the mechanical stress of digestion and absorption, ensuring that the epithelial barrier remains intact. FLNA also participates in intracellular signaling pathways within intestinal cells and regulates cell proliferation and differentiation in the intestinal epithelium [36]. The hyperexpression of FLNA could be a compensatory mechanism for other homeostatic alterations during IBD’s progress, since its upregulation has been observed previously in IBD patients [37].

As seen for FLNA, the expression of moesin (MSN) is significantly upregulated in IBD organoids. MSN is involved in various cellular processes necessary for the correct physiology of the intestine. MSN is a member of the ERM (ezrin–radixin–moesin) protein family. In the context of the intestine, MSN has several important functions, which include the maintenance of the cell structure and the shape of epithelial cells, as it links the actin cytoskeleton to the cell membrane, providing stability and supporting the cellular architecture. This function is critical for the formation and maintenance of the intestinal lining, ensuring its proper barrier function, and is also essential for cell–cell adhesion, cell migration and motility. MSN is also implicated in the inflammation and immune responses in different cell types. In the intestine, it may have a role in regulating the immune response to pathogens and in maintaining the immune homeostasis in the gut.

High mobility group AT-hook 2 (HMGA2) was also expressed at elevated levels in the IBD organoids. HMGA2 contributes to various biological processes, including development, cell proliferation and differentiation. Although HMGA2 is not specific to the intestine, it has been studied in the context of intestinal development and its broader involvement in cell regulation. During embryonic development, HMGA2 is expressed in the intestine and has a role in the formation of the gastrointestinal tract. It contributes to the proper patterning and differentiation of cells, ensuring the development of a functional intestine. HMGA2 participates in regulating cell proliferation, which is critical for tissue growth and repair in the intestine. It helps control the balance between cell division and cell death, ensuring the maintenance of a healthy intestinal epithelium. Stem cells that continuously replenish and renew the lining of the intestinal wall express high levels HMGA2 to properly regulate self-renewal and differentiation [38]. In the intestine, HMGA2 may also contribute to the development of intestinal tumors through its effects on cell proliferation, differentiation and genomic instability. Some studies have suggested that HMGA2 may be indirectly involved in processes that could contribute to intestinal inflammation and IBD pathogenesis [39]. Given the theory that chronic inflammation may serve as a conduit to carcinogenesis through tissue damage and sustained regenerative activities [40], it is plausible to consider proteins such as HMGA2 as potential bridges connecting IBD and cancer.

We employed a network analysis (Figure 4) based on protein–protein interactions and functional information from the String database [41] to visualize the global protein expression in IBD organoids. This comprehensive analysis encompasses both direct (physical) and indirect (functional) associations [41]. To illustrate the impact of IBD on the gene expression of these organoids, we projected our proteomic datasets onto the established protein–protein interaction database for *Canis lupus familiaris*. Remarkably, the proteins that were upregulated and downregulated in IBD organoids exhibited a high degree of interconnectedness (Figure 3). This observation suggests extensive proteome-wide readjustments in response to pathology that could explain the alterations in protein expression associated with functional and physical protein–protein interactions.

The observed differences in the proteome signature between the IBD and control organoids suggest that mutations may have occurred in the IBD organoids because of the chronic inflammation experienced by the donor animals. Chronic inflammation is known to be associated with genomic instability and an increased risk of mutations [42], which could contribute to the altered phenotypic traits and proteomic changes we observed. One way to investigate if mutations are responsible for some of the observed changes and if they have a possible role in disease pathogenesis could be using whole-genome sequencing (WGS) to assess the mutation signature of IBD and identify mutation drivers.

The observed differences in protein expression between the IBD and control organoids also point towards the likelihood of epigenetic changes having a significant role in the development and manifestation of IBD in canine duodenal tissue. Although our research provides valuable insights into the phenotypic traits and proteomic alterations associated with IBD in these organoids, it is essential to acknowledge the potential contribution of epigenetic modifications, which were not directly assessed in this study. To delve deeper into the potential epigenetic changes associated with canine IBD organoids, future investigations could focus on DNA methylation analysis to compare the DNA methylation profiles of IBD and control organoids to reveal differences in gene expression regulation. Another possibility is to study histone modification patterns, including acetylation, methylation and phosphorylation. Chromatin immunoprecipitation assays followed by sequencing (ChIP-seq) can be used for this purpose, among other possibilities, such as non-coding RNA analysis, including small RNA and microRNA profiling [43]. An additional assay that would provide additional information about epigenetic changes is the transposase-accessible chromatin with sequencing (ATAC-Seq) assay, which can determine chromatin accessibility across the genome [44]. Incorporating these approaches into future research endeavors will enhance our understanding of the role of epigenetics in canine IBD and shed light on the complex interplay between the genetic and epigenetic factors in disease pathogenesis.

An intriguing aspect to consider in our study is that the mutations or epigenetic differences observed in the IBD organoids, if any, may come from the initial source of cells used for organoid isolation, which are typically derived from stem cells. The organoid isolation protocol begins with these stem cells, and it is well established that stem cells can accumulate genetic mutations or epigenetic changes over time due to factors such as DNA replication errors, exposure to environmental stressors and chronic inflammation [45]. If this is the case, it would be a satisfactory explanation for the difficulties during the remission of IBD.

In previous studies related to IBD, where the subjects were the dogs that later served as IBD organoid donors in this study, we discovered that both the microbiome and metabolome experienced persistent disruptions even after the remission phase [46,47,48]. In the context of this study, the sustained perturbation of the microbiome and metabolome may suggest that IBD can lead to lasting alterations in the intestinal environment, necessitating more effective interventions such as stem cell therapy.

One notable limitation of our study is the use of control organoids obtained from a single donor (beagle) of a different breed than the IBD donors (Yorkshire terriers). Although these control organoids served as a crucial reference point for our investigation, the genetic and physiological variations between different canine breeds may introduce potential confounding factors. These breed-related differences could impact the baseline characteristics of the control organoids and potentially influence the observed differences between the IBD and control groups. Another significant limitation of our study is the relatively small number of organoids used for the analysis. Although we aimed to draw meaningful conclusions from the available samples, the limited number of donors may not fully capture the heterogeneity of IBD in canine duodenal tissue. This limitation could impact the generalizability of our findings. Further studies involving control organoids from the same breed as the IBD donors would be valuable for confirming and refining our findings. Additionally, future research should consider the influence of breed-specific factors on the observed phenotypic traits and protein alterations in canine duodenal organoids.

In conclusion, our study demonstrates that canine IBD organoids retain several disease-associated traits that were successfully identified through proteome-wide quantitative analysis. We believe that our findings will contribute to the future understanding of canine IBD and highlight the potential of IBD organoids as a valuable in vitro system for studying pathogenicity. This research showcases the applicability of organoids in characterizing chronic enteropathies such as IBD. Furthermore, we have demonstrated the potential of canine IBD to reflect human IBD with respect to biomarkers, suggesting that dogs could serve as a promising preclinical model for investigating novel therapeutic approaches.

## 3. Materials and Methods

### 3.1. Organoid Cultivation

Duodenal samples were obtained from biopsies of three dogs diagnosed with IBD. The healthy control duodenal sample was obtained from a dog euthanized for other reasons unrelated to gastrointestinal disease. The collection of tissue samples was conducted in compliance with the guidelines of the institutional ethics committee and in accordance with the Good Scientific Practice guidelines and Austrian legislation. The use of biopsy tissue material was included in the University’s “owner’s consent for treatment,” which was signed by all patient owners. The isolation of intestinal crypts from the duodenal tissue was performed following established protocols [12,49]. The duodenal tissue section was incubated with 5 mM EDTA (Sigma-Aldrich, St. Louis, MO, USA) for 30 min to dissociate the crypts. Subsequently, the tissue was vigorously shaken until the crypts were released. After two washing steps with PBS and advanced Dulbecco’s modified Eagle’s medium/F12 (DMEM/F12, Invitrogen, Thermo Fisher Scientific, Waltham, MA, USA), approximately 500 crypts were resuspended in 50 µL Matrigel (BD Biosciences, Franklin Lakes, NJ, USA) and seeded per well of a 24-well plate. Following Matrigel polymerization, refined medium was added. The refined medium, containing penicillin/streptomycin, HEPES, GlutaMAX, B27 (with vitamin A), N-acetylcysteine, gastrin, and ALK5 kinase inhibitor (A83-01), was obtained from Tocris Bioscience (Wiesbaden-Nordenstadt, Germany), human hepatocyte growth factor (HGF), human Noggin, human IGF1 and human FGF2 were provided by PeproTech (5 Cedarbrook Drive Cranbury, NJ 08512 USA), as R-sopondin- (Cultrex R-spondin1 cells were obtained from Trevigen, 614 McKinley Place NE Minneapolis, MN 55413, USA) and Wnt3A-conditioned media (L-Wnt3a cells were provided upon request and after material transfer agreement). During the first two days after isolation, the refined medium was supplemented with 50 ng/µL EGF purchased from Thermo Fisher Scientific (Vienna, Austria) and 10 µM ROCK inhibitor (Y-27632, Selleck Chemicals (Houston, TX 77014, USA). Afterward, the medium was changed to non-supplemented refined medium. The growth medium was changed every two to three days. For weekly passaging, organoids were harvested and mechanically disrupted using a flame-polished Pasteur pipette. Depending on the splitting ratio (1:4 to 1:8), the corresponding quantity of organoid fragments was embedded in 50 µL fresh Matrigel, seeded per well of a 24-well plate and cultured with refined growth medium. 

### 3.2. Sample Preparation for Liquid Chromatography–Mass Spectrometry

Five microliters of cell lysate per sample were transferred to a tube containing 20 μL of urea denaturing buffer (6 M urea, 2 M thiourea and 10 mM HEPES; pH 8.0). Disulfide bonds from the cell lysate proteins were reduced by adding 1 μL of dithiothreitol (10 mM) and incubating for 30 min at room temperature. The samples were alkylated by adding 1 μL of iodoacetamide (55 mM) solution and incubated at room temperature for another 30 min in the dark. Four volumes of ammonium bicarbonate buffer (40 mM) were added to each sample, and overnight digestion was carried out at room temperature by adding 1 μg of trypsin protease (Thermo Scientific, Waltham, MA, USA). To stop the digestion reactions, acidification of the samples was achieved by adjusting the final concentrations to 5% acetonitrile and 0.03% trifluoroacetic acid (TFA). Next, the samples were desalted using C18 StageTips with Empore™ C18 Extraction Disks (StageTip format) as previously described [50], and the peptides eluted from the StageTips were dried using vacuum centrifugation.

### 3.3. Sample Analysis

The peptides were dissolved in 40 µL of a solution containing 0.05% TFA and 4% acetonitrile. Thereafter, 1 µL of each sample was applied to an Ultimate 3000 reversed-phase capillary nano liquid chromatography system connected to a Q Exactive HF mass spectrometer (Thermo Fisher Scientific, Waltham, MA, USA). All samples were injected and concentrated on a PepMap100 C18 trap column (3 µm, 100 Å, 75 µm inner diameter (i.d.) × 20 mm, nanoViper; Thermo Scientific) equilibrated with 0.05% TFA in water. After switching the trap column inline, LC separations were performed on an Acclaim PepMap100 C18 capillary column (2 µm, 100 Å, 75 µm i.d. × 500 mm, nanoViper; Thermo Scientific, Waltham, MA, USA) at an eluent flow rate of 300 nL/min. Mobile phase A consisted of 0.1% (*v*/*v*) formic acid in water, whereas mobile phase B contained 0.1% (*v*/*v*) formic acid and 80% (*v*/*v*) acetonitrile in water. The column was pre-equilibrated with 5% mobile phase B, followed by an increase to 44% mobile phase B over 70 min. Mass spectra were acquired in a data-dependent mode, utilizing a single MS survey scan (*m*/*z* 300–1650) with a resolution of 60,000 and MS/MS scans of the 15 most intense precursor ions with a resolution of 15,000. The dynamic exclusion time was set to 20 s and the automatic gain control was set to 3 × 10^6^ and 1 × 10^5^ for the MS and MS/MS scans, respectively.

### 3.4. Data Analysis

The analysis of the MS and MS/MS raw data was conducted using the MaxQuant software package (version 2.0.3.0) with the Andromeda peptide search engine [51]. The data were searched against the *Canis lupus familiaris* reference proteome (ID: UP000002254; downloaded from Uniprot.org, 43,621 sequences) using default parameters, enabling label-free quantification (LFQ) and matching between runs. Data filtering and statistical analysis were performed using Perseus 1.6.14 software [20]. Only proteins identified and quantified with LFQ intensity values in at least two (out of three) replicates were included in downstream analysis. Missing values were imputed from a normal distribution using default settings (width 0.3, downshift 1.8). Mean log2 fold differences between groups were calculated in Perseus using Student’s *t*-test. Proteins with a minimum 2-fold intensity change compared with the control (log2 fold change ≥ 1 or log2 fold change ≤ −1) and a *q*-value (FDR adjusted *p*-values) ≤ 0.05 were considered significantly abundant.

## Figures and Tables

**Figure 1 ijms-25-00576-f001:**
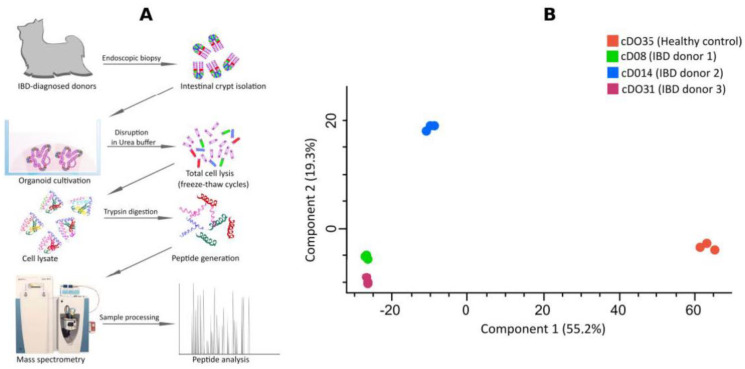
(**A**) Cultivation of duodenal canine organoids from a healthy donor and inflammatory bowel disease (IBD)-derived organoids from three different donors. Differentiated organoids were subjected to label-free quantitative proteomics analysis. (**B**) Principal component analysis (PCA) of quantitative proteome analysis demonstrates the segregation of healthy donor canine duodenal organoids (cDO35) from IBD organoids (cDO8, cDO14 and cDO31) obtained from three different donors and experimental replication consistency (*n* = 3).

**Figure 2 ijms-25-00576-f002:**
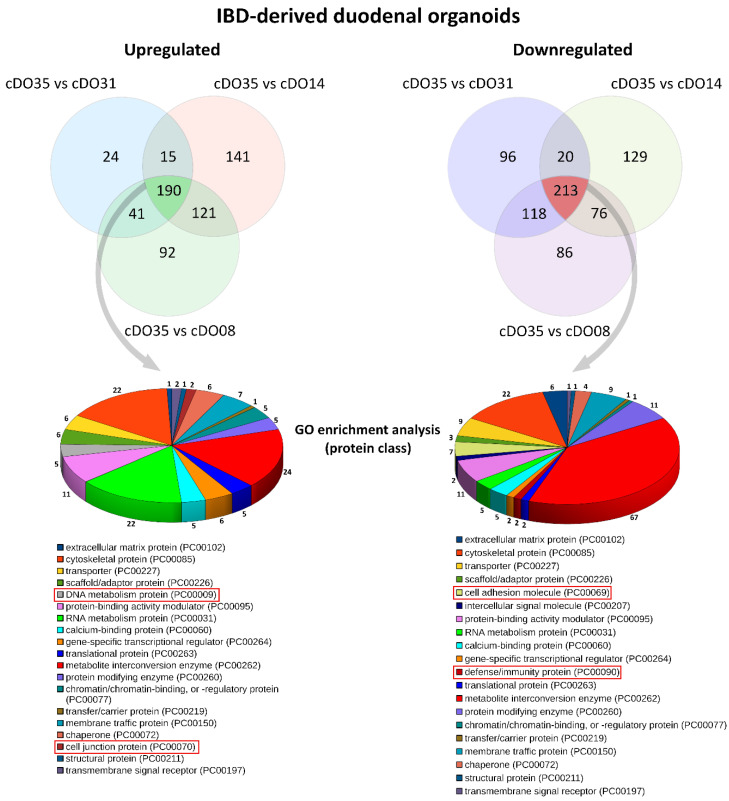
The **top panel** shows Venn diagrams with three intersections representing the significantly up- and downregulated proteins in the comparison of three IBD organoids (cDO08, cDO14 and cDO31) to the control (cDO35). The **bottom panel** provides a visual representation of gene ontology enrichment analysis [26] (for the functional category protein class) of the common proteins that were upregulated (190) and downregulated (213) in the IBD organoids from three independent donors. Only proteins with known functions are represented. The categories that are highlighted with red rectangles were only found in the up or downregulated fractions.

**Figure 3 ijms-25-00576-f003:**
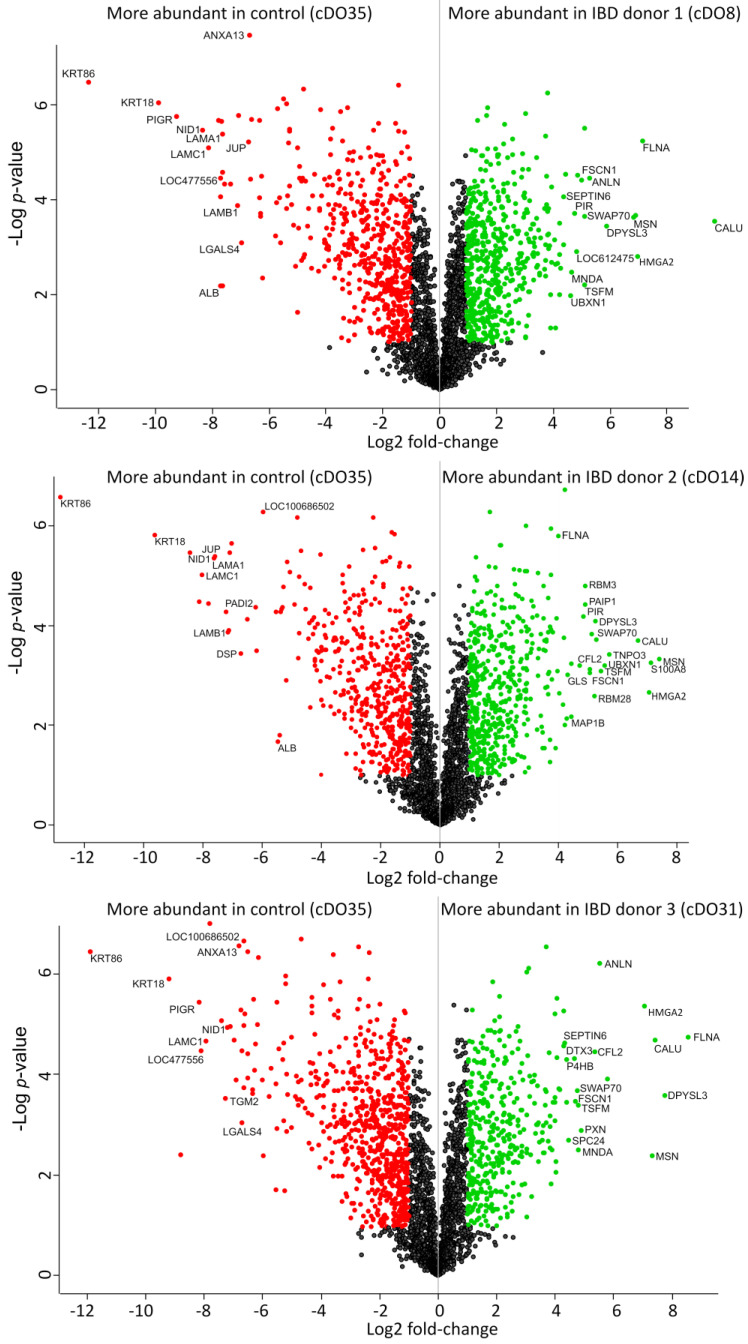
Volcano plots of IBD organoids from three different donors compared with healthy control organoids. In these plots, the log2 fold-change is plotted against the –log of the *p*-value. Protein intensity values are obtained by liquid chromatography-mass spectrometry which are normalized to account for variations in sample loading and other technical biases. The log2 fold change for a given protein is the difference of its logarithm base 2 intensities between the two experimental groups. A positive log2 fold change indicates an upregulation, signifying an increase in protein abundance, whereas a negative value denotes a downregulation, indicating a decrease in abundance. The significantly upregulated (green dots) and downregulated (red dots) proteins are indicated in every plot (adjusted *p*-value < 0.05 and log2 fold change ≥ 1 or ≤ −1). Non-significant differentially expressed proteins are represented by black dots. Only selected proteins from the top up- and downregulated fractions are labelled.

**Figure 4 ijms-25-00576-f004:**
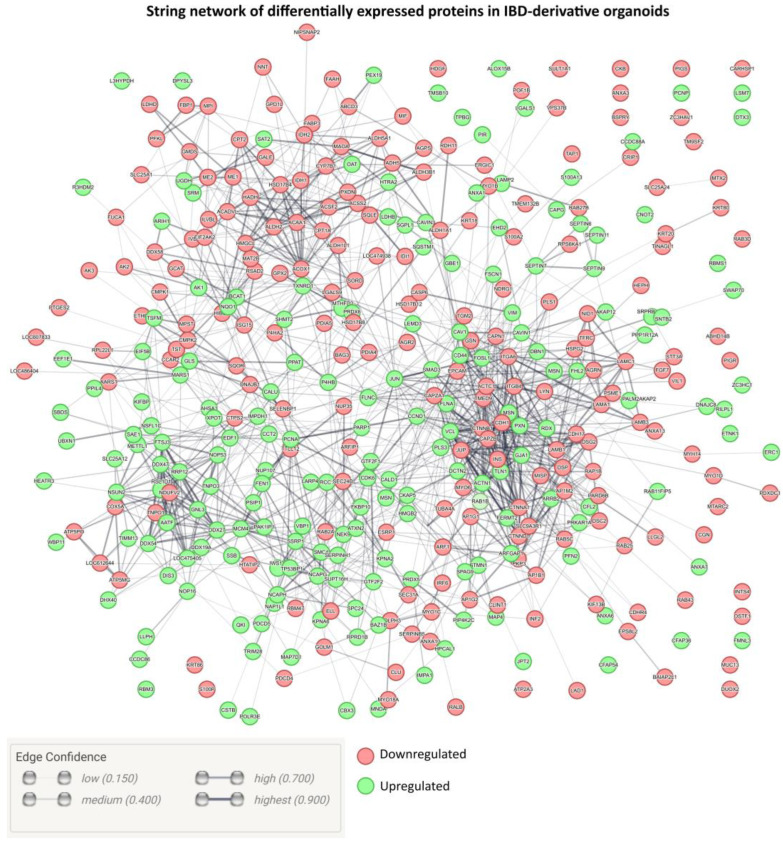
Network visualization of differentially expressed proteins in IBD organoids [41]. In the graph, green nodes signify upregulated proteins, whereas red nodes depict downregulated ones. Additionally, the edge thickness reflects the statistical confidence of the interactions. Gene names were used instead of protein names.

## Data Availability

All data are freely available under request.

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
