# Peer review of "Traces of Canine Inflammatory Bowel Disease Reflected by Intestinal Organoids"

_ijms, 2024, doi:10.3390/ijms25010576_

Round 1

Reviewer 1 Report

Comments and Suggestions for Authors

This paper investigates Inflammatory Bowel Disease (IBD) from duodenal organoids of canine individuals, attempting to determine if there are phenotypic differences between the organoids of healthy and IBD patients. In order to analyze the entire proteome, organoids were subjected to quantitative proteomics via mass spectrometry. The findings are conclusive, and there are significant differences between the two groups.

1) It appears that only three IBD donors and one healthy control were used? Please be more explicit about that in the text.

2) The data and figures are not insufficiently explained.

2a) Figure 1 depicts component 1 and component 2 (I understand that a Principal Component Analysis is being performed), but it is not adequately explained where these components originate.

2b) Figure 3 lacks sufficient information: the volcano plots represent log(p) vs log(fold-change), but it is unclear how to calculate the fold-changes whose logarithm appears on the horizontal axis.

2c) In general, once exposed, statistical analyses appear conclusive, but it is unclear what measurements they were based on.

Comments on the Quality of English Language

English is quite acceptable

Author Response

Review 1. This paper investigates Inflammatory Bowel Disease (IBD) from duodenal organoids of canine individuals, attempting to determine if there are phenotypic differences between the organoids of healthy and IBD patients. In order to analyze the entire proteome, organoids were subjected to quantitative proteomics via mass spectrometry. The findings are conclusive, and there are significant differences between the two groups.
1) It appears that only three IBD donors and one healthy control were used? Please be more explicit about that in the text. R. Thank you for your comment. This has been emphasized in the limitations of our study in the third paragraph of the Results and Discussion. 2) The data and figures are not insufficiently explained. R. Thank you for your comment. We have expanded the explanations of the figures and data. 2a) Figure 1 depicts component 1 and component 2 (I understand that a Principal Component Analysis is being performed), but it is not adequately explained where these components originate. R. We understand your concern, and for that reason, we have added additional explanation and a reference on how PCA is used for multidimensional OMIC data in the Results and Discussion section, as follows:
Perseus incorporates principal component analysis (PCA) based on singular value decomposition (SVD) [19], a computational approach well-suited for high-dimensional data. PCA within Perseus identifies the primary effects in the data and reveals the proteins responsible for the separation of different proteomic states [24].
19. Alter O, Brown PO, Botstein D. Singular value decomposition for genome-Wide expression data processing and modeling. Proc Natl Acad Sci U S A. 2000;97: 10101–10106. doi:10.1073/PNAS.97.18.10101/ASSET/FF30CA99-28C8-4A76-B3E7-5915430E79B5/ASSETS/GRAPHIC/PQ1702748006.JPEG
24. Tyanova S, Temu T, Sinitcyn P, Carlson A, Hein MY, Geiger T, et al. The Perseus computational platform for comprehensive analysis of (prote)omics data. Nat Methods. 2016;13: 731–40. doi:10.1038/nmeth.3901 2b) Figure 3 lacks sufficient information: the volcano plots represent log(p) vs log(fold-change), but it is unclear how to calculate the fold-changes whose logarithm appears on the horizontal axis. R. Thank you for your comment. In order to enhance readability and comprehension, we have made modifications to Figure 3 of the volcano plot. Additionally, we've provided further clarification in the legend, which now reads as follows: "The raw intensity values for each peptide or protein are obtained from mass spectrometry data. These values have been normalized to account for variations in sample loading and other technical biases. The log2 fold-change for a given protein is calculated by taking the base-2 logarithm of the ratio of its abundance in one condition to another. A positive log2 fold-change indicates an upregulation, signifying an increase in protein abundance in the first condition, while a negative value denotes a downregulation, indicating a decrease in abundance." 2c) In general, once exposed, statistical analyses appear conclusive, but it is unclear what measurements they were based on.
R. Thank you for your inquiry. Label-free quantification (LFQ) in proteomics is a method for comparing protein abundance across diverse experimental conditions, eliminating the need for stable isotopic labels. To conduct statistical analyses on LFQ data, researchers employ a range of measurements and statistical techniques to identify differentially expressed proteins and evaluate result reliability. The quantification is typically based on the UV absorption of proteins detected by the mass spectrometer, a well-established procedure in proteomics. While this step is considered standard and widely recognized within the field, detailed explanations are often omitted in manuscripts due to its established nature. Citations are routinely provided for those seeking additional information on this widely accepted and established practice in proteomic analysis.

Reviewer 2 Report

Comments and Suggestions for Authors

Author’s explored the potential derivation and use of canine intestinal organoids as a model for the study of the efficacy of therapeutic interventions, by means of proteomic analysis.

The paper is of intrest and explore a new approach towards personalized medicine. I have no specific concerns about the paper which is well written and explore also the pros/cons of the study

Results and discussion

Please provide figures 1 and 2, I can see only the legends.

Comments on the Quality of English Language

General comments

Please have a review of spelling and typing.

Author Response

Reviewer 2. Author’s explored the potential derivation and use of canine intestinal organoids as a model for the study of the efficacy of therapeutic interventions, by means of proteomic analysis. The paper is of intrest and explore a new approach towards personalized medicine. I have no specific concerns about the paper which is well written and explore also the pros/cons of the study Results and discussion Please provide figures 1 and 2, I can see only the legends.
R. Thank you for your feedback. We acknowledge and regret the visibility issue with Figures 1 and 2 in the PDF version. We have corrected this issue in the version, and we appreciate your patience. Additionally, we have thoroughly reviewed the manuscript, addressing any identified typos and ensuring consistent language use for enhanced clarity and coherence. Your comments have been invaluable in refining our work.

Round 2

Reviewer 1 Report

Comments and Suggestions for Authors

The clarification on the PCA seems to me to be very insufficient, a technicality relating to the Singular Value Decomposition in the Perseus software is cited. This detail is well known to any reader who knows the PCA technique, but really with such a cryptic and convoluted explanation, I am left with severe doubts if the authors understand the limitations underlying a PCA and its objective green.

Similar with the volcano plot, a less frequent type of technique and of which the article makes instrumental use, without providing too many clarifications about its function. The explanations seem to have been written by someone who doesn't really fully understand the details of that type of chart and has just used software that just provides the chart. I think it would have been worthwhile to clarify why it was decided to use this type of graph and how its use could clearly show a scientifically valid conclusion.

Overall, I find that the authors have done little to clarify the article, and I still get the impression that the article has obvious explanatory gaps.

Author Response

Thank you for sharing your concerns regarding our article. We appreciate your critical evaluation of our work, and we take your feedback seriously. It's crucial for us to address your points and improve the clarity and transparency of our content.

  1. PCA Clarification: Your observation about the insufficiency of our PCA explanation, particularly the reference to the Singular Value Decomposition in the Perseus software, has been duly noted. We understand your frustration with what seems like a cryptic and convoluted explanation. We apologize for any confusion and in the new version we have added additional information and additional citations about PCA. Our goal was to ensure that the methodology was presented in a more accessible manner, with a focus on eliminating unnecessary technical jargon. In proteomics, the PCA relies on feature abundance levels across samples to ascertain the principal axes of abundance variation. We can effectively segregate run samples based on abundance variations by transforming and visualizing the abundance data within the principal component space. This approach is also valuable for pinpointing run outliers and enhancing the identification process.

In other words, PCA is a widely employed dimensionality reduction technique, particularly in managing large datasets. This method involves transforming a comprehensive set of variables into a more concise one, retaining most of the information from the original dataset. This process effectively streamlines complex data structures while preserving the essential information inherent in the larger set.

Principal components generate new variables crafted through linear combinations or mixtures of the original variables (in our case, the number of identified proteins, and intensities of the UV absorption by protein peptides in the mass spectrometer detector (what is called label-free intensities). This combination is orchestrated to ensure that the newly formed variables, termed principal components, exhibit no correlation. Moreover, most of the information inherent in the initial variables is condensed or compressed into the primary components. To illustrate, in a 5-dimensional dataset, PCA endeavours to allocate the maximum conceivable information into the first component, followed by the maximum remaining information into the second component, and so forth.

It's crucial to recognize that the principal components lack inherent interpretability and genuine meaning. This is because they are constructed as linear combinations of the original variables, resulting in representations that may not directly convey intuitive or concrete information. This is why it looks so cryptic.

Certainly, we haven't included this information in the manuscript in such detail because no one in the field of proteomics typically does so, unless they are developing a new method of analysis. However, this is not the focus of our article. Instead, we are utilizing a tool created by other researchers, which has been thoroughly validated and is appropriately applied in this manuscript.

  1. Volcano Plot Usage: We appreciate your concern regarding the perceived lack of clarity regarding our choice to utilize a volcano plot and its contribution to scientifically valid conclusions. However, we respectfully disagree with the assertion that a volcano plot is a less frequently used technique. Contrary to this view, a volcano plot is widely acknowledged as one of the most commonly employed tools for representing omics data. Its popularity stems from its ability to encapsulate the two pivotal variables in expression data: the fold-change (depicted here as the log2 fold-change, where log2 serves as a scale transformation for grouping data and creating symmetry) and the significance of expression data through the -log of the p-value. This approach enables the representation of all detected proteins in a single plot, with significant ones highlighted and the most relevant ones labelled. Such a practice is considered standard in proteomics.

Notably, the Perseus computational platform, utilized for generating this data, has been cited in 6043 articles, with a majority incorporating volcano plots. To underscore the prevalence of volcano plots, consider the reference (https://www.sciencedirect.com/science/article/pii/S1476927113000169) illustrating their common use, not only in proteomics but also in representing mRNA expression levels. In this case, we believe there isn't much room to provide additional explanation about something that is entirely standard and widely used.